# A Multivariate Meta-Analysis for Optimizing Cell Counts When Using the Mechanical Processing of Lipoaspirate for Regenerative Applications

**DOI:** 10.3390/pharmaceutics15122737

**Published:** 2023-12-06

**Authors:** Gershon Zinger, Nia Kepes, Ron Kenett, Amos Peyser, Racheli Sharon-Gabbay

**Affiliations:** 1Hand Unit, Department of Orthopedic Surgery, Shaare Zedek Medical Center, Faculty of Medicine, Hebrew University of Jerusalem, Jerusalem 91120, Israel; rachelish@szmc.org.il; 2Department of Neuroscience, Michigan State University Lyman Briggs College, East Lansing, MI 48824, USA; kepesnia@msu.edu; 3The KPA Group, Ra’anana 4353701, Israel; ron@kpa-group.com; 4The Samuel Neaman Institute, Technion, Haifa 3200003, Israel; 5Department of Orthopedic Surgery, Shaare Zedek Medical Center, Faculty of Medicine, Hebrew University of Jerusalem, Jerusalem 91120, Israel; amosp@szmc.org.il

**Keywords:** lipoaspirate, mechanical processing, meta-analysis, multivariate, optimization, JMP^®^

## Abstract

Lipoaspirate has become the preferred source for regenerative cells. The mechanical processing of lipoaspirate has advantages over enzymatic processing but has a lower yield of regenerative cells. A review of the literature shows different techniques of extraction, but the ideal method or combination has not been determined. Methods: A comprehensive literature search was focused on the mechanical processing of lipoaspirate, without the use of enzymes. Data from the articles were integrated by utilizing a multivariate meta-analysis approach and used to create a statistical-based predictive model for a combination of multiple variables. Results: Starting with 10,000 titles, 159 articles were reviewed, and 6 met the criteria for inclusion and exclusion. The six studies included data on 117 patients. Sixteen factors were analyzed and six were identified as significant. The predictive profilers indicated that the optimal combination to maximize the cell yield was: a centrifuge force of 2000× *g*, a centrifuge time of 10 min, a cannula diameter of 2 mm, and an intra-syringe number of passes of 30. The optimal patient factors were a higher BMI and younger age. Conclusions: The novelty of the method used here was in combining data across different studies to understand the effect of the individual factors and in the optimization of their combination for mechanical lipoaspirate processing.

## 1. Introduction

Adipose tissue is an abundant and easily accessible source of mesenchymal stem cells (MSCs) [1]. MSCs have the potential to differentiate into a variety of cell types and play a significant role in tissue regeneration [2]. The quality and quantity of MSCs obtained from lipoaspirate are highly dependent on the processing method used. Enzymatic methods using collagenase produce the greatest yield of regenerative cells [3]. However, enzymatic methods have disadvantages that include regulatory restrictions (FDA guidance, 2014b,c), the cost, and the time needed for processing [3]. Many mechanical methods have been proposed for lipoaspirate processing [4], but the optimal method or combination of methods remain unclear.

The yield of MSCs from lipoaspirate is affected by many processing factors. Understanding the effect of the different processing methods on the MSCs yield is crucial for the design of a multi-factor intervention technique. In this study, a set of statistical tools was used that combined evidence from published studies, in an approach that resembles a meta-analysis [5]. In general, a meta-analysis typically aims at deriving aggregate measures such as odds ratios, which is considered controversial by the medical community [6]. In a traditional meta-analysis, effect sizes from studies that address the same research question are pooled together. Advances in multivariate meta-analysis can estimate outcome-specific effects for multiple outcomes [7] and for multiple factors that synthesize available factors simultaneously [8].

A multi-variate meta-analysis using JMP^®^ software (Version 16 Pro) was used to create a model that predicts the ideal combination of methods to optimize the highest cell count for the mechanical processing of lipoaspirate.

## 2. Materials and Methods

### 2.1. Study Design

A comprehensive literature search was conducted using PRISMA guidelines [9] on PubMed, from 1 January 2000 to 26 April 2023. The search was performed using the search phrase: “isolation” or “method” or “mechanical” or “dissociation” AND (“lipoaspirate” or “adipose”). The inclusion criteria included studies published in English that evaluated mechanical methods of subcutaneous lipoaspirate processing and reported outcome data that included the nucleated cell count. The exclusion criteria included articles that evaluated enzymatic methods for either processing or for counting, review articles, primary animal studies, cell counts after culturing, sources other than subcutaneous, and those reporting only clinical outcomes (Figure 1). The senior author (GZ) performed the initial title screening. Two investigators (RS and NK) independently reviewed all abstracts. The goal at this screening stage was to be inclusive of any potential studies that met the inclusion criteria. After removing studies that, from the abstracts, met the exclusion criteria, full texts of the remaining studies were retrieved. The full texts were assessed based on the same inclusion and exclusion criteria, but the detail allowed for a more accurate screening.

### 2.2. Data Extraction and Outcome Normalization

The data were extracted into a spreadsheet, with the factors identified during the articles review (Table 1). This included patient characteristics such as age and sex (Table 2), surgeon-chosen factors such as the cannula size and liposuction site, and the methods used for mechanical processing (Table 3). The outcomes measure (response) was the nucleated cell count concentration. These data were independently extracted (RS and NK) and then compared. Disagreement was resolved by consensus. All data were then reviewed to confirm accuracy (RS and GZ).

To enable the comparison between the different studies, the cell count results were normalized relative to the volume of the initial liposuction aspirate, as shown in Table 4. In some cases, this was calculated from the information provided in the text. If the starting lipoaspirate volume was unavailable, the corresponding author was contacted and in all cases was able to provide these critical data to allow for a comparison between the studies.

In addition, the data included the statistical methods used to analyze the results, the sample size, the type and value of results, and the error and the *p*-values, which were extracted into the summary tables.

### 2.3. Multivariate Meta-Analysis Involving Multiple Factors

#### 2.3.1. Impact of Missing Data

The presence of missing data negatively influences the fitting of statistical models. Corresponding authors were contacted to provide any missing data. For some factors, some information remained unavailable. Specifically, only one study exhibited some missing data, while five studies provided a complete set of the factors (Table 3). Prior to conducting the statistical analysis, we applied an “Automated Data Imputation” feature using JMP^®^ software (version 16 Pro, SAS Institute (Cary, NC, USA) for the remaining missing data. This method utilized the mean values of the factors to impute the missing data points.

#### 2.3.2. Result Standardization

In this study, the primary focus was on assessing and optimizing the yield of nucleated cells obtained through the mechanical processing of lipoaspirate, referred to here as the response variable (y). As demonstrated in Table 4, the reported results are presented in various units, making direct comparisons challenging. To facilitate the comparison of the cell count yields across different articles (Table 1) and to conduct statistical analyses, the data were standardized. The standardization process consisted of calculating normalized cell counts, which represent the nucleated cell count yield relative to the initial lipoaspirate volume (Table 4).

#### 2.3.3. Multivariate Analysis

A multivariate regression analysis was conducted using JMP^®^ software, with the response variable (y) representing the normalized yield of nucleated cells per mL of starting lipoaspirate obtained through the mechanical processing. The results obtained from the literature search were used as the experimental array. If, for example, two mechanical methods were used in the same article, these two methods were considered as separate in the experimental array. The 16 experimental factors (x) are listed in Table 5. The sample size of each study was used in JMP^®^ as the modeling frequency.

Two distinct models, the Standard Least Squares and the Mixed models, were fitted and assessed for their goodness of fit. The Least Squares model is used to determine the best-fitting line or curve for a set of data points. The objective is to minimize the sum of the squared vertical distances between the actual data points and the corresponding points predicted by the model. This entails adjusting the parameters of a chosen model (such as the slope and intercept of a line) to minimize the sum of squared differences. Widely employed in regression analysis, this method is instrumental in revealing the most accurate relationship between variables, particularly when dealing with data variability or noise. The Least Squares model is limited by its sensitivity to outliers and the reliance on sufficient data [16].

A Mixed model, or mixed-effects model, is an extension of linear models that incorporates both fixed effects and random effects. The Mixed model is particularly useful when dealing with nested or repeated measurements. While Mixed models offer flexibility, they present challenges. They require a careful balance between fixed and random effects, and the estimation process can be computationally intensive. Additionally, assumptions about the random effects’ distribution need consideration, and determining the appropriate level of complexity can require experience. Despite these challenges, mixed models are valuable for handling diverse data structures, providing a more realistic representation of complex relationships [17].

The Standard Least Squares model included variance analysis, an assessment of factors’ importance, and the identification of critical factors. The Mixed model incorporates random effects to account for variability across the different studies and fixed effects related to the factors under investigation (Table 5). The Mixed model includes random effects covariance analysis with respect to the article name and fixed effects analysis for the factors: BMI, age, centrifuge time, centrifuge force, cannula diameter, and intra-syringe pass count. When considering the effect of the study factor on the output variable, critical factors are defined as those that exhibited a *p* value < 0.001. Important factors are defined as those with 0.001 < *p* < 0.05. Factors that exhibited an importance score with a *p*-value > 0.05 were considered as not important and excluded from the model.

For the Mixed model, a conditional residual quantile plot, a useful diagnostic tool in regression analysis, was generated for assessing the model’s fit. In addition, an Actual by Predicted plot was also made to visually assess the performance of the predictive model. The visual assessment is an effective tool for comparing the actual (observed) values of the dependent variable with the predicted values generated by the model.

For both the Standard Least Squares and the Mixed models, the Prediction Profiler was used to predict the performance of factor combinations that were not used in any of the studies reviewed. The Predictive Profiler was used to identify the combination of factors that maximized the outcome measure—in this case, the nucleated cell count concentration. The JMP^®^ Profiler enables the simultaneous optimization of multiple factors for the output variable while considering the impact of noise on the analysis [18].

#### 2.3.4. Setup-Related Bias

The JMP^®^ Design Evaluation is a tool that is used to determine whether the experiment is designed properly. This tool evaluates if there are sufficient data spread over the experimental array to build a reliable model [19]. In this study, the JMP^®^ Design Evaluation tool was used to quantify the bias (percent error) after the model was created. To visualize the study performance (percent reliability), a Fraction of Design Space plot was employed. Furthermore, the models’ performance was assessed through power and prediction variances [20]. These are post hoc evaluations and have limited interpretation.

## 3. Results

The systematic review yielded 12,598 references (Figure 1). PubMed truncates after 10,000 titles. Therefore, only the first 10,000 titles were manually screened, and 396 abstracts were evaluated. None of the last 2800 titles of the 10,000 screened were already relevant. After a review of the complete abstracts, or if the abstract was unavailable, full articles were reviewed. Additional articles were identified during the review process. A total of 159 articles were chosen for full text review by two independent reviewers. From these, only six met the criteria and were included in this systematic review and analysis (Table 1). These six articles included data on 117 volunteers. These six articles were mined for data extraction and for subsequent analysis. Of the six articles, two articles, Cicione [10] and Chaput [11], each used two separate mechanical methods for processing. Therefore, these two articles were counted as four experiments in the array. Overall, the array included eight experiments used in the multivariate regression analysis. The patient characteristics are summarized in Table 2, and the mechanical factors are summarized in Table 3. A summary of the normalized cell counts is given in Table 4.

### 3.1. Multivariate Regression Analysis

In this study, 16 factors that could affect the nucleated cell count concentration were evaluated (Table 5). Multivariate regression analysis was used to study the association between the factors and the cell count. Two different models were fitted: Standard Least Squares and Mixed models. From the initial 16 factors, 6 were found to be significant (*p* < 0.05) and included in the regression analysis (Figure 1). In addition, all six factors were identified as critical (*p* < 0.001): two patient-related and four mechanical factors.

Figure 2 and Figure 3 are the Actual by Predicted plots that visually demonstrate the performance of both models, the Standard Least Squares and Mixed models, respectively. Each data point, on these plots, corresponds to an observation from the data presented in Table 2 and Table 3. For the Standard Least Squares model (Figure 2), all the data points align along the 45-degree diagonal red line, signifying that the actual values match the predicted values, indicating a high model accuracy (R^2^ = 0.97, *p* value < 0.001). For the Mixed model, the evaluation of the pattern and deviation of the data points in Figure 3 shows that the Mixed model’s predictions do not align optimally with the actual outcomes. This departure from the diagonal line suggests the presence of potential systematic errors or biases in the model’s predictions. These discrepancies may be linked to the limited dataset and random effects stemming from variations between articles.

In addition, the Mixed model’s performance was also assessed through the distribution of the residuals (the difference between the actual and the predicted value; Figure 4), with constant variance across different levels of the factors. The residuals exhibited a consistent pattern across different factors. This suggests that the Mixed model adequately captured the relationship between the factors and the response variable for those conditions. The spread of residual points around the diagonal line was consistent along the line, confirming homoscedasticity (constant variance of residuals). These findings confirm that the Mixed model exhibited good suitability and effectiveness.

### 3.2. Setup-Related Bias

While the Evaluation of Study Design is typically applied in the context of DOE ahead of the data collection, in this study, its purpose was to evaluate the viability of conducting this multivariate meta-analysis with just six articles (eight experiments). The power analysis yielded a *p*-value of less than 0.05, and the expected root mean squared error (RMSE) was estimated to be 1. When assessing the study design using the Fraction of Design Space plot (Figure 5), it revealed an 80% accuracy probability for the predictive models. In other words, there is a 20% chance that the model is biased and will not accurately predict the optimal mechanical combination.

### 3.3. Maximizing the Cell Count in Lipoaspirate Processing

To forecast the ideal combination of factors for maximizing the cell count, predictive profilers were generated based on the Least Squares and Mixed models. An example of a profiler setting is presented in Figure 6 and Figure 7, respectively. In these figures, the predicted model is represented by the black line, while the blue lines depict the statistical confidence limits. The vertical red dashed line indicates the setting of the factor value that, when considered in conjunction with other factors, shows the predicted cell count shown by the horizontal red dashed line. For example, in Figure 6, the setting of 10 intra-syringe passes, in combination with the other factor settings, predicts a cell count of 76,937 cells per 1 mL of lipoaspirate. Employing the maximize-desirability function, the optimized setup for achieving the maximum cell count per 1 mL of lipoaspirate is summarized in Table 6. Both models agree regarding the optimal factor values. With the ideal combination of factors (Table 6), the Standard Least Squares model predicts a cell count of 158,631 cells per mL of lipoaspirate, compared to the Mixed model, which predicts 167,529 cells per mL of lipoaspirate, a difference of only 5.3%.

## 4. Discussion

Mechanical processing offers many advantages over enzymatic methods, including a reduced time and cost and fewer regulatory restrictions. Each of the published articles identified in this systematic review used a different combination of factors for mechanical processing. In this study, a novel method using JMP^®^ software was used to integrate the data from the different studies to create a predictive model that optimizes mechanical processing. The goal for optimization is to maximize the concentration of nucleated cells, an indirect measure of the number of regenerative cells.

The isolation methods of pluripotential mesenchymal cells from adipose tissue at the point of care have direct applications in many fields such as plastic surgery, orthopedic surgery and regenerative medicine. Aronowitz et al. [3] performed a review that compared published articles that used both mechanical and enzymatic methods for processing lipoaspirate. The gold standard remains using proteolytic enzymes to break down the extracellular matrix, releasing the cells into the solution and giving the highest concentration of nucleated cells. In contrast, the stromal vascular fraction (SVF) obtained through mechanical techniques is safe, avoids regulatory restrictions, is more cost-effective, and is less time-consuming. Mechanical methods have some benefits but have a lower yield when compared to enzymatic methods [3]. In this study, we performed a systematic literature review and a multifactorial meta-analysis to identify the best combination of mechanical processing methods to improve the concentration of nucleated cells.

The number of studies that used only mechanical methods of lipoaspirate processing to yield ASCs without the addition of an enzyme is notably scarce. We considered including articles that used mechanical methods for processing but used an enzyme for cell counting but decided that including these studies would give erroneous results. Comparing two mechanical methods where one study used an enzyme for cell counting might theoretically consider one mechanical method as more superior when it was the counting that was more efficient.

This study had two goals. The primary goal was to identify the optimum method of the mechanical processing of lipoaspirate based on the current published data. The second goal was to demonstrate the ability of combining the data from different studies to create a predictive model that would identify this ideal combination. For that purpose, the JMP^®^ statistical software provided a powerful dynamic tool that identified the effect of each factor on the cell count and provided the ideal combination of factors. A similar analysis could be used in a future study to optimize for other outcome measures such as specific phenotypes, cell types, or the differentiation potential. In addition, a similar method could be used to study MSC processing from other sources such as bone marrow.

The initial step in the analysis required determining the important factors to include in the predictive model (Figure 1). Of the 16 factors, only 6 were statistically important. However, the other 12 factors might also have an impact on other outcome measures. For example, although the source of lipoaspirate was not found to be a statistically critical or important factor for maximizing cell count, it might have an impact on the effectiveness of a specific application. For example, some lipoaspirate sources might be more effective at reducing joint inflammation, and other lipoaspirate sources might be more effective at healing diabetic wounds.

The next step after choosing the factors to analyze is to create the predictive model. Other models were considered before choosing the Standards Least Squares and Mixed models. However, they were not included in the analysis based on a lack of fit using parameters such as R^2^, *p*-values, and Bayesian Information Criteria (BIC).

The validation of modeling is essential for the multivariate meta-analysis of multiple factors. When the effect of factors is large compared with the number of collected studies, the estimated variance–covariance matrix using the frequentist method may be inconsistent [21], leading to poor estimates. The plot showed that 80% of the predictive models were accurate, meaning they provided reliable predictions. However, the remaining 20% showed a high level of variability, indicating that they were not consistent or reliable in making predictions. This variability reduces the confidence in the accuracy of these particular predictive models. To improve the accuracy of the predictive model, more data would need to be added to the model. However, we were unable to find additional published studies that met our criteria. In the future, as more studies are published on mechanical processing, data may be added to the study array, which will reduce the setup-related bias in the analysis.

The predictive profiler was instrumental in visualizing the specific effects of each factor concurrently. Since this feature is dynamic, the images seen in Figure 3 and Figure 8 show examples of one combination of factor settings. Table 6 summarizes the predicted setup factors for maximizing the cell yield. Both the Least Squares and Mixed models yielded the same predicted combinations, which increases the confidence in the methodology. The next phase of this investigation will be to test the predicted optimum combination using Quality by Design methodology.

While conducting a meta-analysis on a set of articles, an intriguing anomaly surfaced through the residual plot—a data point that conspicuously deviated from the anticipated pattern of residual distribution (Figure 8). This particular data point has raised questions about its consistency with the rest of the dataset. Notably, this outlier corresponds to a study that advocates for the use of a specific product, a factor that could potentially impact the reliability and integrity of the results. It is imperative to underscore the significance of result reliability when performing a meta-analysis, as the inclusion of data points influenced by external factors may introduce bias and affect the overall robustness of the analysis. In order to enable the generation of the models (minimal data points), it was included. Removing this outlier would have improved the model reliability. However, the improved outcome may be the result of a technical innovation and therefore needed to be included.

One of the limitations of this study was the lack of uniformity in how the different authors presented their data. For example, one study expressed the absolute cell count and another expressed the concentration. Some studies reported the initial lipoaspirate volume while others reported the volume after decanting. For future comparative analysis, we suggest that specific details are included in future studies (Table 7). This standardization would greatly facilitate the comparison of studies.

In the course of this systematic review, out of a total of 10,000 articles evaluated, only six met the criteria for this study. Although the dataset comprises only six articles, there were 117 volunteers. While this is an admittedly limited number for robust statistical analysis, this scarcity underscores the importance of including specific data factors that would allow for the integration of data.

## 5. Conclusions

In a traditional meta-analysis, data from different studies are used to analyze the association of a single variable with the outcome measure. The novelty of the method used here was in combining data across different studies to understand the effect of the individual factors and in the optimization of their combination for mechanical lipoaspirate processing. This was carried out by using a powerful program to create predictive models based on least fit criteria. The statistical method revealed which factors are unimportant and identified the crucial factors along with their optimal settings. For achieving a maximum cell count, the following factors are suggested to maximize the nucleated cell count concentration: a centrifuge speed at 2000× *g* for 10 min, a 2 mm cannula diameter, and 30 passes through the intra-syringe.

## Data Availability

Additional data are available upon request.

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
