# Peer review of "A Multivariate Meta-Analysis for Optimizing Cell Counts When Using the Mechanical Processing of Lipoaspirate for Regenerative Applications"

_pharmaceutics, 2023, doi:10.3390/pharmaceutics15122737_

Round 1

Reviewer 1 Report

Comments and Suggestions for Authors

The authors of the article demonstrate the use of a statistical model for understanding the factors that influence the cell count using mechanical processing of lipoaspirate for regenerative medicine purposes. The factors that were found to influence outcomes of mechanically processed lipoaspirate include age, sex, lipoaspirate volume, final volume of final product and BMI. The statistical model helped to identify critical factors in the processing of lipoaspirates.

The authors need to consider the following points,

1.       In the studies evaluated, were the cell surface markers evaluated in these studies and how did the processing affect the quality of the lipoaspirate ? It is known that enzymatic digestion can influence surface markers and so does mechanical processing assists in reducing this loss.

2.       Site of lipoaspirate was found not to be a critical indicator for optimizing cell counts. However, the site could influence the subsequent use of application and its differentiation potential, e.g. Hoffa’s fat pad ASCs for knee cartilage ? Does your model need to include a focus on the final lipoaspirate application and not solely the cell count to show its efficiency for the desired treatment ? Has

3.       Has the model been tested for other similar types of processing (e.g. bone marrow aspirate isolation for mesenchymal stromal cells) to see the most efficient and high yield processes for the isolation of other specific cell types ? This should be presented by the authors to further validate the model.

Author Response

A multivariate meta-analysis for optimizing cell count when using mechanical

processing of lipoaspirate for regenerative applications

Manuscript ID: pharmaceutics-2726993

Reviewer Question

Response

Change in manuscript

Reviewer 1:

1. The authors of the article demonstrate the use of a statistical model for understanding the factors that influence the cell count using mechanical processing of lipoaspirate for regenerative medicine purposes. The factors that were found to influence outcomes of mechanically processed lipoaspirate include age, sex, lipoaspirate volume, final volume of final product and BMI. The statistical model helped to identify critical factors in the processing of lipoaspirates.

Agreed

No change

1. In the studies evaluated, were the cell surface markers evaluated in these studies and how did the processing affect the quality of the lipoaspirate ? It is known that enzymatic digestion can influence surface markers and so does mechanical processing assists in reducing this loss?

Your question addresses quality and the study addresses quantity. In all six studies, cell surface markers were evaluated. Theoretically, they could be added as a second parameter, in addition to cell count, to perform a multivariate meta-analysis. However, the focus of the study was to present a statistical methodology that could be used to predict a combination of mechanical methods to optimize a specific outcome. To address the question of whether mechanical processing helps reduce the impact of enzymatic effects on cell surface markers, a comparative study could be conducted.

Added lines 399-402:

A similar analysis could be used in a future study to optimize for other outcome measures such as specific phenotypes, cell types, or differentiation potential.

2. Site of lipoaspirate was found not to be a critical indicator for optimizing cell counts. However, the site could influence the subsequent use of application and its differentiation potential, e.g. Hoffa’s fat pad ASCs for knee cartilage ? Does your model need to include a focus on the final lipoaspirate application and not solely the cell count to show its efficiency for the desired treatment?

It is certainly possible that the site of lipo-harvesting may have a different impact on the effectiveness of ultimate application. However, this clinical question cannot be answered with the articles we chose that were focused on maximizing cell count. A similar statistical technique could be used to measure clinical outcomes based on lipoaspirate source. However, this would require a separate literature review with inclusion and exclusion criteria to include lipoaspirate source and a common clinical outcome measure. Then a meta-analysis with this combination could be performed if there was sufficient data.

Added Lines: 406-410:

However, the other 12 factors might also have an impact on other outcome measures. For example, although the source of lipoaspirate was not found to be a statistically critical or an important factor for maximizing cell count, it might have an impact on the effectiveness on a specific application. For example, some lipoaspirate sources might be more effective at reducing joint inflammation and other lipoaspirate sources at healing diabetic wounds.

3.  Has the model been tested for other similar types of processing (e.g. bone marrow aspirate isolation for mesenchymal stromal cells) to see the most efficient and high yield processes for the isolation of other specific cell types? This should be presented by the authors to further validate the model

It  The inclusion criteria in this study was only the subcutaneous source. We purposely did not include articles describing bone marrow mechanical processing. In lipoaspirate, the mechanical processing attempts to separate MSCs from the adipose tissue. Your questions emphasize the need of using our approach to model data for different applications. Future models would benefit from incorporating a broader scope that includes the final lipoaspirate application and other cell sources to optimize both quantity and quality of the outcome product. This could include establishing a predictive model for cell surface markers, along with conducting various characterization tests such as viability, colony-forming unit (CFU) analysis, adhesion, and differentiation potential. In this study, we demonstrated the application of multivariate meta-analysis using JMP® software for optimizing a single parameter. This was done to simplify the understanding of this complex technique while recognizing the need for a comprehensive approach to cell characterization.

Added Lines 401-402:

In addition, a similar method could be used to study MSC processing from other sources such as bone marrow.

Reviewer 2 Report

Comments and Suggestions for Authors

The submitted manuscript investigates the optimization of the highest cell count for mechanical processing of lipoaspirate by performing a multivariate meta-analysis using JMP® software. The article is concise and clear. The article has the following comments:

1. In the introduction, describe the validity and reliability of multivariate meta-analysis using JMP® software.

2. Explain whether there are any limitations to applying multivariate meta-analysis.

3. In the conclusion, present the conclusions of a multivariate meta-analysis on optimizing cell count when using mechanical processing of lipoaspirates for regenerative applications.

Comments on the Quality of English Language

The article needs minor revision for language and grammar.

Author Response

A multivariate meta-analysis for optimizing cell count when using mechanical

processing of lipoaspirate for regenerative applications

Manuscript ID: pharmaceutics-2726993

Reviewer Question

Response

Change in manuscript

Reviewer 2:

In the introduction, describe the validity and reliability of multivariate meta-analysis using JMP® software.

JMP is a general-purpose statistical software platform with extensive use in biostatistics. In the paper we test its modeling capabilities to assess the effects of covariates using data combined from different studies. Additional information was added about the statistical methodology. More information about the limitation of the fitted models was added to lines 129-148. In addition, models' performance was assessed through power and prediction variances.

Added Lines 129-146:

The Least Squares model is used to determine the best-fitting line or curve for a set of data points. The objective is to minimize the sum of the squared vertical distances between the actual data points and the corresponding points predicted by the model. This entails adjusting the parameters of a chosen model (such as the slope and intercept of a line) to minimize the sum of squared differences. Widely employed in regression analysis, this method is instrumental in revealing the most accurate relationship between variables, particularly when dealing with data variability or noise. The Least Squares model is limited by its sensitivity to outliers and the reliance on sufficient data [10].

A Mixed model, or mixed-effects model, is an extension of linear models that in-corporates both fixed effects and random effects. The Mixed model is particularly useful when dealing with nested or repeated measurements. While Mixed models offer flexi-bility, they present challenges. They require a careful balance between fixed and random effects, and the estimation process can be computationally intensive. Additionally, as-sumptions about the random effects' distribution need consideration, and determining the appropriate level of complexity can require experience. Despite these challenges, mixed models are valuable for handling diverse data structures, providing a more realistic representation of complex relationships [11].

2. Explain whether there are any limitations to applying multivariate meta-analysis.

There is real potential for disparities between the prediction models and the tested values due to two main issues: 1. Insufficient data and 2. Unreliable data. We discuss these limitations in lines 417-428 quantifying the reliability of the model. Specifically, given that the data was based on only 6 references, the model quantifies its own reliability, here at 80%.

3. In the conclusion, present the conclusions of a multivariate meta-analysis on optimizing cell count when using mechanical processing of lipoaspirates for regenerative applications.

The conclusions were added as suggested as well as the addition of Table 6.

Added Lines 519-522:

For achieving a maximum cell count, the following factors are suggested to maximize nucleated cell count concentration: centrifuge speed at 2000 g for 10 minutes, a 2 mm cannula diameter, and 30 passes through the intra-syringe.

Round 2

Reviewer 1 Report

Comments and Suggestions for Authors

The authors have addressed my points appropriately.